# Genome-Wide Identification of GYF-Domain Encoding Genes in Three *Brassica* Species and Their Expression Responding to *Sclerotinia sclerotiorum* in *Brassica napus*

**DOI:** 10.3390/genes14010224

**Published:** 2023-01-15

**Authors:** Xiaobo Zhang, Lei Qin, Junxing Lu, Yunong Xia, Xianyu Tang, Xun Lu, Shitou Xia

**Affiliations:** 1Hunan Provincial Key Laboratory of Phytohormones and Growth Development, College of Bioscience and Biotechnology, Hunan Agricultural University, Changsha 410128, China; 2College of Life Science, Chongqing Normal University, Chongqing 400047, China; 3Agricultural Science Academy of Xiangxi Tujia and Miao Autonomous Prefecture, Xiangxi 416000, China

**Keywords:** *B. napus*, GYF domain, synteny analysis, *S. sclerotiorum*, infection response

## Abstract

GYF (glycine-tyrosine-phenylalanine)-domain-containing proteins, which were reported to participate in many aspects of biological processes in yeast and animals, are highly conserved adaptor proteins existing in almost all eukaryotes. Our previous study revealed that GYF protein MUSE11/EXA1 is involved in nucleotide-binding leucine-rich repeat (NLR) receptor-mediated defense in *Arabidopsis thaliana*. However, the GYF-domain encoding homologous genes are still not clear in other plants. Here, we performed genome-wide identification of *GYF-domain encoding genes* (*GYFs*) from *Brassica napus* and its parental species, *Brassica rapa* and *Brassica oleracea*. As a result, 26 GYFs of *B. napus* (*BnaGYFs*), 11 *GYFs* of *B. rapa* (*BraGYFs*), and *14 GYFs* of *B. oleracea* (*BolGYFs*) together with 10 *A. thaliana* (*AtGYF*s) were identified, respectively. We, then, conducted gene structure, motif, cis-acting elements, duplication, chromosome localization, and phylogenetic analysis of these genes. Gene structure analysis indicated the diversity of the exon numbers of these genes. We found that the defense and stress responsiveness element existed in 23 genes and also identified 10 motifs in these GYF proteins. Chromosome localization exhibited a similar distribution of *BnaGYFs* with *BraGYFs* or *BolGYFs* in their respective genomes. The phylogenetic and gene collinearity analysis showed the evolutionary conservation of *GYFs* among *B. napus* and its parental species as well as *Arabidopsis*. These 61 identified GYF domain proteins can be classified into seven groups according to their sequence similarity. Expression of *BnaGYFs* induced by *Sclerotinia sclerotiorum* provided five highly upregulated genes and five highly downregulated genes, which might be candidates for further research of plant–fungal interaction in *B. napus*.

## 1. Introduction

The glycine-tyrosine-phenylalanine (GYF) domain, which was originally identified in CD2BP2 as an adaptor domain responsible for binding to the PPG motif of the T cell adhesion molecule CD2 [1], was defined as a small, versatile adaptor-conserved domain that recognizes proline-rich sequences (PRS) [2]. Along with more GYF domain-containing proteins reported, they were classified approximately into CD2BP2-type and SMY2-type based on their different binding models [2]. It was suggested that GYF domains are involved in translation initiation, mRNA splicing, supervision, and repression of translation, ubiquitin ligation, signal transduction, regulation of immune protease, and proper wound healing response in animals [3,4,5,6,7,8].

There also exists a set of plant-specific GYF domain-containing proteins [9]. In *Arabidopsis thaliana*, Needed for RDR2-independent DNA methylation (NERD) was reported to play a role in chromatin-based RNA silencing [10]. Mutant, snc1-enhancing 11 (MUSE11)/Essential for poteXvirus accumulation 1 (EXA1) is responsible for loss-of-susceptibility to plantago asiatica mosaic virus and is involved in nucleotide-binding leucine-rich repeat (NLR) receptor-mediated defense [11,12]. Plant SMY2-type ILE-GYF domain-containing protein 1 (PSIG1) dampens the induction of cell death during plant–pathogen interactions [13]. *AT1G24300* and *AT1G27430* are close homologs of *AtEXA1*; all of them showed significant enrichment when copurified with TUTase URT1, suggesting that they function in siRNA biogenesis [14]. However, unlike *AtEXA1*, both *AT1G24300* and *AT1G27430* do not have similar impact on plant defense [12]. Under X-ray treatment, phosphorylated *AT1G24300* showed increased abundance, which means it might participate in DNA damage response [15]. 

The genus *Brassica* species provide important vegetable and oilseed crops cultivated worldwide. Chinese cabbage (*Brassica rapa* L.), broccoli (*Brassica oleracea* L.), and rapeseed (*Brassica napus* L.) are mostly cultivated as vegetables for human consumption and for producing oil, condiments, and fodder because of their nutrient contents, including vitamins, proteins, and minerals (e.g., zinc, iron, sodium, and potassium) [16,17,18]. However, oil rapeseed has been harassed from a bunch of diseases; the most concerning pathogen is *S. sclerotiorum*, which generates stem rot and causes serious damage to the yield and quality of rapeseed [19], resulting in massive losses annually. More than 400 plant species have suffered from *S. sclerotiorum* [20,21]. *S. sclerotiorum* produces cellulase, pectinase, and cutinase after infection of the host, decomposing cell wall polymers and disrupting the structural integrity of the wall [22,23]. *S. sclerotiorum* also secretes oxalic acid, phospholipase, and proteolytic enzymes to weaken the host’s defense and provide rich nutrients for itself when it invades the plant [24]. Thus, control of rapeseed disease has long been a widely concerning issue. In order to survive in disease events, plants have evolved a precise defense system against pathogens along with the long coevolution history [25]. *B. napus* (AACC), the main planted type of rapeseed crop, is formed by the natural hybridization of *B. rapa* (AA) with *B. oleracea* (CC) [26]. The AA or CC subgenome of *B. napus* shares homologies with its respective parental genomes, as well as *A. thaliana* [27]. 

Previously, *Bra*027983 was identified as a SET domain containing protein [28], while *Bra*037238 and *Bra*037299 were identified as plant homeodomain (PHD) finger proteins and upregulated under drought and salt treatment [29]. In addition, *BnaA09g16090D* was found to be the target of sRNAs which were upregulated during *S. sclerotiorum* infection [30]. Recently, GYF protein *St*EXA1 was reported as facilitating potato virus Y (PVY) accumulation in potatoes through the SG-dependent RNA regulatory pathway [31]. Nevertheless, the GYF-domain encoding homologous genes are still not clear in *Brassica* plants. We, then, conducted a genome wide identification of GYF-domain encoding genes (*GYFs)* in *Brassica* species, and analyzed the expression pattern of all *GYFs* in *B. napus* (*BnaGYFs*) under the induction of *S. sclerotiorum*, which might provide an understanding of plant–pathogen interaction profiling of GYF proteins.

## 2. Materials and Methods

### 2.1. Identification of GYFs

The generic feature file (GFF), FASTA DNA and protein sequence alignment file (FNA) of *B. napus*, *B. rapa*, *B. oleracea*, and *A. thaliana* were downloaded from BRAD (http://Brassicadb.cn/, accessed on 13 January 2023) [32], Ensemblplants (https://plants.ensembl.org/index.html, accessed on 13 January 2023), and the TAIR (http://www.arabidopsis.org, accessed on 13 January 2023), respectively. Through protein family code, family name, and BLAST search as well as HMM confirmation, 61 *GYF*s were finally confirmed.

### 2.2. Gene Structure, Motif Predicjtion, Cis-Element Prediction, Sequence Alignment, and Phylogenetic Analysis

After we downloaded the genomic sequences and CDS sequences of *BnaGYFs* from Ensemblplants (https://plants.ensembl.org/index.html, accessed on 13 January 2023), like in former studies [33], GSDS2.0 was recruited to draw the gene structure of *BnaGYFs*, and MEME suite [34] was recruited for the motif prediction of *BnaGYFs*. Similar to former studies [35], we conducted cis-element prediction, the promoter regions (about 2000-bp upstream) of *BnaGYFs* were extracted from genome general feature format (GFF) files using PlantCARE (http://bioinformatics.psb.ugent.be/webtools/plantcare/html/, accessed on 13 January 2023), and visualized by TBtools [36]. Sequence alignments by Muscle, and a neighbor-joining (NJ) tree was generated with the alignments using MEGA 11.0 software with bootstrap analysis from 5000 replicates. The phylogenetic tree was modified by Adobe Illustrator. Upset plots for motifs and cis-elements were generated on SRplot website (http://www.bioinformatics.com.cn/en?keywords=upsetR, accessed on 13 January 2023).

### 2.3. Synteny Analysis and Chromosome Distribution Visualization of GYFs

Syntenic analysis and the duplication of *Brassica GYFs* and *AtGYFs* were calculated using the MCScanX program in TBtools [36,37]. Chromosome distribution information of *Brassica GYFs* was extracted from GFF file. TBtools [36] was recruited to visualize the chromosome distribution and the tandem duplications of *Brassica GYFs*.

### 2.4. Plant Growth and Infection Treatment

Seeds of *B. napus* variety Zhongshuang11 were provided by Zhongsong Liu (Hunan Agricultural University, Changsha 410128, China). Plants were grown in a growth room at 20–22 °C, with a 16 h light and 8 h dark photoperiod, 30 d seedlings were recruited for infection treatments. *S. sclerotiorum* strain 1980 was obtained from Jeffrey Rollins (University of Florida). *S. sclerotiorum* was grown in a chamber at 20–22 °C on PDA plates, and 2-day-old fungus was recruited for the infection treatments. Leaves of 30 d seedlings were inoculated with mycelia, harvested after 0, 24, 48, and 72 h post inoculation, flash-frozen in liquid nitrogen, and stored at −80 °C. Three plants with the closest phenotype and growth status were harvested, and the experiment was repeated three independent times.

### 2.5. RNA Extract and RT-qPCR

For the expression analysis of *BnaGYFs*, total RNA of the 0, 24, 48, and 72 h postinoculation was isolated from the infected leaves using the Eastep™ Super Total RNA Extraction Kit (Promega, Madison, WI, USA). Reverse transcription was carried out using the GoScript™ Reverse Transcription System (Promega, Beijing, China). The RT-qPCR assay was carried out using 2 × SYBR Green Premix Pro Taq HS Premix (AG11702, Accurate Biotechnology (Hunan) Co., Ltd., Changsha, China)) and a Step-One real-time fluorescence PCR instrument (Applied Biosystems, Bedford, MA, USA). The RT-qPCR reaction system contained 10 ng cDNA, 4 µM of each primer, 5 µL 2 × SYBR Green Premix Pro Taq HS Premix, 0.2 µL ROX reference dye, and 3.4 µL RNAase-free water. The RT-qPCR programming was as follows: denaturation at 95 °C for 120 s, followed by 40 amplification cycles (95 °C for 20 s, 55 °C for 20 s, and 72 °C for 30 s). *BnaActin7* was used as an internal housekeeping gene. Two or more independent biological replicates and three technical replicates of each sample were performed for quantitative PCR analysis. Gene-specific primers used in the experiments are listed in Appendix A.

### 2.6. RT-qPCR Results Analysis of BnaGYFs Induced by S. Sclerotiorum

The results of RT-qPCR were analyzed using Excel 2018. The internal reference gene is *BnaACTIN7*. Relative gene expression levels were analyzed using the 2^−ΔΔCT^ method [38]. The RNA-seq data (Accession Number: GSE81545) was downloaded from NCBI [39]. The heatmaps were drawn by TBtools [36].

## 3. Results

### 3.1. Identification and Motif Analysis of GYFs

When searching with the protein family of PF02213 in the Ensemblplants database, 17 GYF domain-encoding genes were obtained, but more genes were found when GYF was searched in the Ensemblplants database. We also used all known GYF sequences as queries to perform a BLAST against BRAD and the Ensemblplants database, then the Hidden Markov Model (HMM) was used for further confirmation of the candidate *GYFs*. As a result, 26 GYF genes were confirmed in *B. napus* (AACC). In addition, 11 in *B. rapa* (AA), 14 in *B. oleracea* (CC), 10 in *A. thaliana* were identified by similar processes. Then, we analyzed the gene structure of all *BnaGYFs*, the exon numbers showed great diversity, ranging from 3 (*BnaA09g08740D*) to 21 (*BnaA09g09480D*, *BnaA01g18730D*, and *BnaC06g11340D*), and *BnaA01g18730D* showed similar gene structure features with *BnaC06g11340D* (Appendix A). While *BnaA03g41160D* has the longest intron (6231 bp) among *BnaGYFs*, *BnaA02g32860D* (3135 bp) has the second long one.

Then, we analyzed the conserved motifs in *Bna*GYFs, the length of *Bna*GYFs ranges from 460 (*BnaC07g03700D*) to 2298 aa (*BnaA01g18730D*) (Appendix A). To better understand the motifs of GYFs in *Brassicaceae*, we generated 10 motifs from GYF proteins of *Brassica napus*, *B. rapa* (AA), *B. oleracea* and *A. thaliana* by HMMER analysis, 61 GYF proteins were analyzed, of them, 23 contain only GYF motif, 8 contain all 10 motifs. Motif2 exists in 33 genes, which is the second most common motif among all analyzed GYF proteins (Figure 1).

### 3.2. Chromosomal Localization and Collinearity Analysis of GYFs

Among the 26 *BnaGYFs* identified, 13 of them were mapped onto chrA (Figure 2A) and the other 13 were mapped onto chrC (Figure 2B). Only 13 of the 21 chromosomes contain *BnaGYFs*. ChrA09 contains five *BnaGYFs* (the highest), chrC09 contains four (the second). However, *BnaGYFs* are absent on eight chromosomes (Figure 2). Only three tandem duplication events were found, including *BnaC03g57590* with *BnaC03g57610* on chrC03 in *B. napus*, *Bra010950* with *Bra010951* on A08 in *B. rapa*, and *Bo3g141450* with *Bo3g141470* on C3 in *B. oleracea*. Similar to *B. napus*, chrA09 contains five *BraGYFs* (the most) in *B. rapa*, and chrC9 contains four *BolGYFs* (the most) in *B. oleracea* (Figure 2). In general, *BnaGYF*s distribute on A or C subgenome in a similar way to *BraGYFs* or *BolGYFs* in their respective genomes.

Subsequently, the duplications of 26 *BnaGYF* genes were analyzed by MCScanX [37] and visualized by TBtools [36], and the results showed 10 *BnaGYF* gene pairs in *B. napus* with only 2 in *A. thaliana* (Figure 3). When the syntenic relationship of *GYFs* between *A. thaliana* and *Brassica* genomes was investigated, we observed 11 gene pairs between *A. thaliana* and *B. rapa*, 18 between *A. thaliana* and *B. oleracea*, 25 between *B. rapa* and *B. napus*, and 48 between *B. oleracea* and *B. napus* (Appendix A). Ka/Ks analysis was also performed by TBtools [36] between *A. thaliana* and *B. napus*, and *B. rapa* and *B. oleracea*, respectively. The Ka/Ks value of all pairs was <0.5 (Appendix A), suggesting that the main force for the evolution of those *GYF* gene pairs was negative selection.

### 3.3. Phylogenetic and Promoter Cis-Acting Elements Analysis of GYFs

To clarify the evolutionary relationship between *Brassica GYFs* and *AtGYFs*, we performed multiple sequence alignment and phylogenetic tree construction of GYF family members in *B. rapa*, *B. oleracea*, *B. napus*, and *A. thaliana*. A total of 61 GYF proteins from 4 species were classified to seven groups (Figure 4). Group I contained 15 proteins (the most), group VII contained 13 (the second), group III contained only 4 genes (the least). Sequence alignment showed the conserved signature polymorphism of each group (Appendix A). Additionally, the sequence logo showed the conserved signature of GYF domain in *Brassica* GYFs and *At*GYFs (Appendix A).

Upstream 2000 bp of *Brassica GYFs* and *AtGYFs* were extracted for cis-acting elements analysis (Figure 5). Several stress-related cis-acting elements were analyzed, including anaerobic induction, abscisic acid (ABA) responsiveness, low-temperature responsiveness, salicylic acid (SA) responsiveness, gibberellin responsiveness, auxin responsiveness, zein metabolism regulation, defense and stress responsiveness, and wound-responsive element. Most genes (54 genes) have the anaerobic induction element, 43 genes have the abscisic acid-responsiveness element (the second), 33 genes have the gibberellin-responsiveness element (the third). The defense and stress responsiveness element exists in 23 genes, while the wound-responsive element only exists in 5 genes (the least). *BnaA09g16220D* contains 18 cis-acting elements (the most); however, *BnaC09g09630D* contains only 2 defense and stress responsiveness elements and 1 low temperature-responsiveness element (the least).

### 3.4. Expression of BnaGYFs under S. Sclerotiorum Induction

As a well-known crop pathogen, *S. sclerotiorum* has caused great agricultural loss all over the world. As shown in cis-element analysis, 38.4% of *BnaGYFs* possess the defense and stress responsiveness element (Appendix A), and some GYF proteins were reported to be involved in plant defense [11,12,13]. Thus, we investigated the expression pattern of *BnaGYFs* under the stress of *S. sclerotiorum*. Tween-six *BnaGYF* genes were identified in this study, we then tried to test the expression for all of them. As a proper real-time primer pair for *BnaC09g16860D* failed, we tested the expression of the 25 *BnaGYF* genes at 24, 48, and 72 h postinoculation of *B. napus* cultivar ZHONGSHUANG11 (ZS11). Five genes were significantly upregulated (fold change > 2) when challenged with *S. sclerotiorum* and were nine significantly downregulated (fold change < 0.5). *BnaA01g18730* and *BnaA09g09480D* were upregulated significantly (fold change > 5). *BnaA02g32860D*, *BnaC02g41610D*, *BnaA09g54650D*, *BnaC03g57610D*, and *BnaC03g00170D* were downregulated significantly (fold change ≤ 0.25) (Figure 6A and Appendix A).

We also searched the expression profiling data (Accession Number: GSE81545) of *GYFs* from the GEO database, 24 hpi (hour postinoculation) expression data of Westar and Zhongyou821 (ZY821) were presented [39]. From these downloaded data, only 16 *BnaGYFs* have valid expression data; we then compared the expression data of the 16 *BnaGYFs* with our data in ZS11 (Figure 6B). Most genes of the 16 genes showed a similar trend when challenged with *S. sclerotiorum*. Expression of 8 *GYFs* were upregulated 24 hpi, 5 genes were downregulated 24 hpi in all three *B. napus* cultivars. The other three *GYFs* were slightly downregulated in ZS11, while they were distinctly upregulated in Westar and ZY821. Different cultivars usually possess different resistance against pathogens; spatiotemporal differences might also be the reason. *GYFs* were obviously upregulated more in ZY821 than in Westar, this explained why ZY821 is a noticed mid-resistant cultivar, while Westar is a low-resistant cultivar [40].

## 4. Discussion

As a multifunctional domain, the GYF domain has been reported to be involved in plant defense in recent years [11,12,13]. Since *B. napus (*AACC) is a natural hybridized species by two parental species, *B. rapa* (AA) and *B. oleracea* (CC) [26], we identified 26 *GYF* genes in *B. napus* (AACC), 10 in *A. thaliana*, 11 in *B. rapa* (AA), and 14 in *B. oleracea* (CC) in this study. By confirming the GYF domain in these proteins with MEME suite, the *GYF domain-encoding genes* (*GYFs*) in these species were obtained. *B. napus* contains a higher number of *GYFs* than the total of its parental species, *B. rapa* and *B. oleracea*, and three *Brassica* species also contain a higher number *GYFs* than that of *A. thaliana*. According to U’s triangle theory, the triplication of ancestor *Brassiceae* genomes and allopolyploidization between its two parent species might have led to the expansion of the *BnaGYF* gene family in *B. napus* [27]. Gene structure analysis showed the diversity of *BnaGYFs*, some have many and long introns, while others only have few short introns (Appendix A). Generally, increased number and length of introns might provide an advantage for alternative splicing and functional diversity [41]. Chromosome localization showed the similarity between the AA subgenome and the CC subgenome of *B. napus* with *B. rapa* (AA) and *B. oleracea* (CC) (Figure 2). Ten *BnaGYFs* contain cis-acting elements of defense and stress responsiveness, only three of them showed upregulated expression after *S. sclerotiorum* infection; seven of them were downregulated. *BnaA01g18730D* contains two copies of defense and stress responsiveness cis-acting elements and its expression was upregulated the most at 72 h post-inoculation with *S. sclerotiorum*. However, *BnaC06g11340D* contains four copies which are the most, yet its expression was suppressed after *S. sclerotiorum* infection. Gene expression has a complex set of regulatory mechanisms, whereas cis-acting elements play only a minor role. Ten gene pairs of *BnaGYFs* were identified by MCScanX; among them, six pairs showed similar expression when challenged with *S. sclerotiorum*, while the other four pairs behaved differently. *BnaA01g18730D*, *BnaC06g11340D*, and *BnaA05g13380D* were together classified to group III, but the expression of *BnaC06g11340D* with the other two genes showed opposite results after *S. sclerotiorum* infection. *BnaC06g11340D* and *BnaA05g13380D* are gene pairs, and they share 65.68% sequence similarity, they both contain only GYF motif among the 10 motifs that we predicted with MEME suite (Appendix A), but they showed different cis-acting element composition (Appendix A). Tandem and segmental duplication events might contribute to the evolution and amplification of gene families [42]. The most common segmental duplication event in plants produces additional family members on different chromosomes [43]. Gene duplications and translocations after hybridization are common in natural hybridized species where inaccurate assembly universally occurred during genetic recombination; it might be the reason of the expression differences within gene pairs [44].

*BnaA09g09480D* and *BnaA01g18730D* upregulated the most among *BnaGYFs*; the former is classified to group II in the phylogenetic analysis, while the latter belongs to group III. *BnaA01g32860D* and *BnaC02g41610D* are gene pairs, and they share 76.7% sequence similarity, and are both downregulated extremely after *S. sclerotiorum* infection. *BnaA01g32860D* and *BnaC02g41610D* were classified to group IV in the phylogenetic analysis, and we found that *BnaA01g32860D* has an ortholog *Bra029374* in *B. rapa* (AA), and *BnaC02g41610D* has an ortholog *Bo2g160890* in *B. oleracea* (CC). In the phylogenetic analysis, *Bna*A09g16220D is the nearest homolog of *At*EXA1; they share 66.2% sequence similarity, and they were both classified in group VII. However, in cultivar ZS11 and Westar, the expression of *BnaA09g16220D* was almost unchanged after *S. sclerotiorum* infection, only in cultivar ZY821, it was slightly upregulated. *At*EXA1 was identified as a plant GYF domain protein and reported to be involved in translational repression of R protein and is indispensable for PlAMV infection in *A. thaliana*, which plays a negative role in plant defense [11,12]. Additionally, expression of *At5g42950* (*AtEXA1*) was upregulated after *S. sclerotiorum* infection in former research [45]. Function deterioration might take place during the evolution. *GYFs* obviously upregulated more in ZY821 than in Westar, because ZY821 is a noticed mid-resistant cultivar, while Westar is a low-resistant cultivar. Different cultivars usually possess different resistance abilities toward pathogens; spatiotemporal differences might also be the reason for the difference [40]. Taken together, we found that most of *BnaGYFs* responded to *S. sclerotiorum* infection in *B. napus*, some were upregulated and some were downregulated. *BnaA09g09480D*, *BnaA01g18730D*, *BnaA05g13380D*, *BnaC07g03700D*, and *BnaA10g18060D* were greatly upregulated after *S. sclerotiorum* infection. Whereas *BnaA01g32860D*, *BnaC02g41610D*, *BnaA09g54650D*, *BnaC03g57610D*, and *BnaC03g00170D* were greatly downregulated after *S. sclerotiorum* infection, which might provide good candidates for further functional analysis.

## 5. Conclusions

We performed a genome wide identification of *Brassica GYFs* and *AtGYFs*, and their gene structure analysis, motif prediction, cis-acting element prediction, chromosome localization, phylogenetic analysis, and expression analysis were conducted in this study. As a well-known domain, the GYF domain showed conservatism between *A. thaliana* and *Brassica* species, and the phylogenetic analysis and collinearity analysis provided valuable insight on the evolutionary characteristics of *Brassica GYFs*. The phylogenetic analysis also provided clues for further function excavation of *Brassicaceae GYFs* involved in drought and salt response as well as siRNA biogenesis. Our results provided an insight into the evolution of GYF domain proteins in *Brassica* species and provided clues for further investigations of the function of *Brassicaceae* GYFs in development and immune response, and the *S. sclerotiorum* responsive genes identified in this study could be useful for further comprehensive study on plant–pathogen interactions and molecular breeding of disease-resistant rapeseed.

## Figures and Tables

**Figure 1 genes-14-00224-f001:**
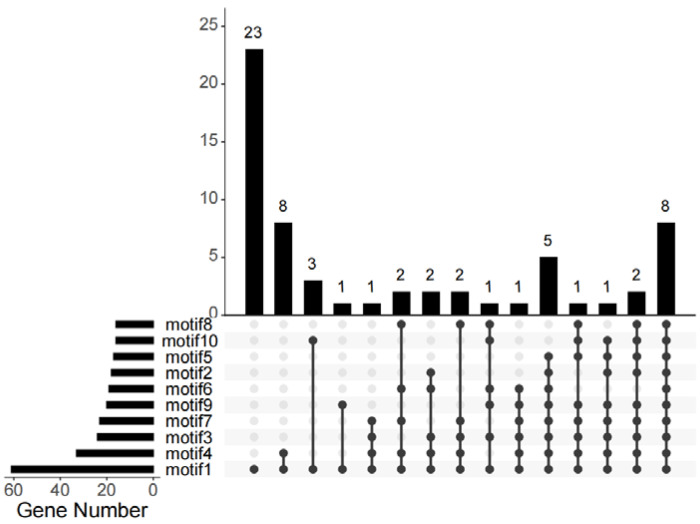
Distributions of motifs in *Brassica GYFs* and *AtGYFs*. The UpSet plot shows the distribution of motifs in *Brassica GYFs* and *AtGYFs*. The number chart above represents the number of genes contained in each type of *GYF*s. The bar chart at the bottom left represents the number of events included in each type of motif. The dotted line shows the type of motifs contained in the group.

**Figure 2 genes-14-00224-f002:**
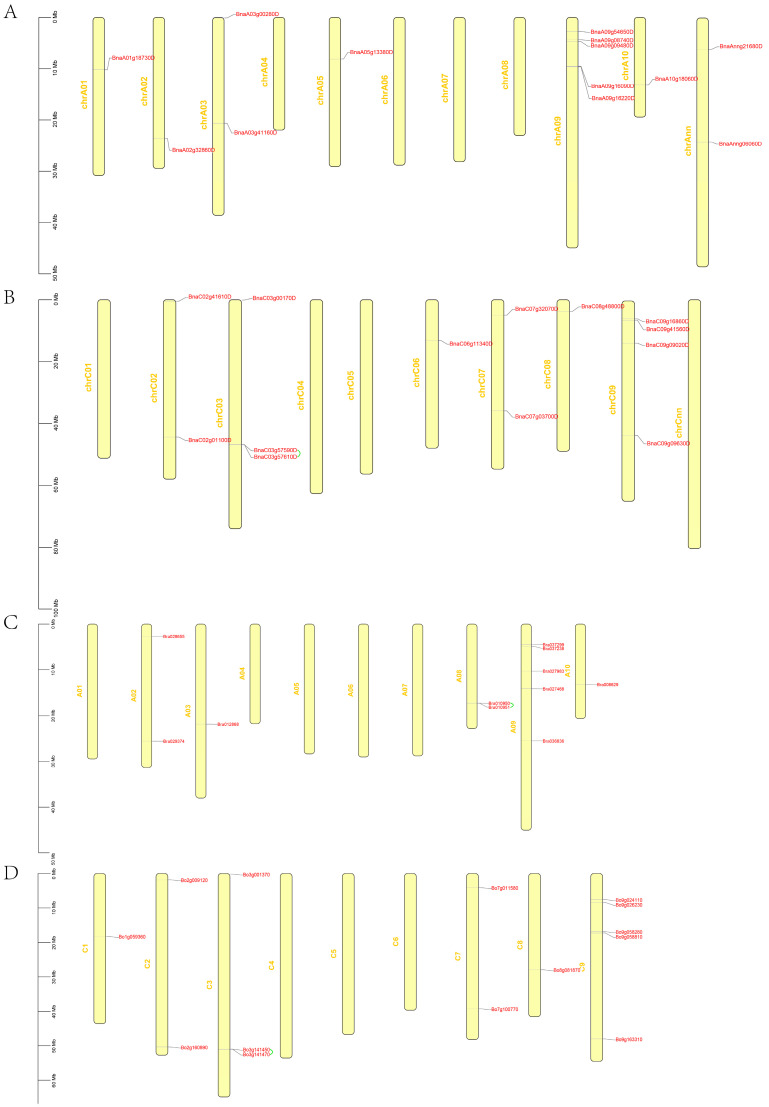
Locations of *Brassica GYFs* on chromosomes. (**A**,**B**) Locations of *GYF*s on A or C subgenome in *Brassica napus*. (**C**) Locations of *GYFs* on chromosomes in *Brassica rapa*. (**D**) Locations of *GYFs* on chromosomes in *Brassica oleracea*. A01-A09, C1-C9, and chrA01-chrCnn represent the chromosome number in *B. rapa*, *B. oleracea*, and *B. napus*, respectively. The green line represents tandem duplication.

**Figure 3 genes-14-00224-f003:**
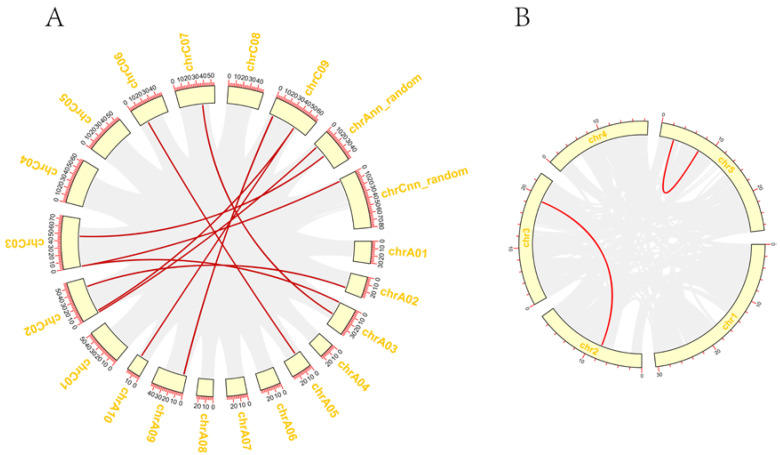
Duplication of *BnaGYFs* and *AtGYFs*. (**A**) Duplication of *GYFs* in *B. napus*. (**B**) Duplication of *GYFs* in *Arabidopsis thaliana*. The red line represents the gene pairs, and the yellow boxes represent the chromosomes.

**Figure 4 genes-14-00224-f004:**
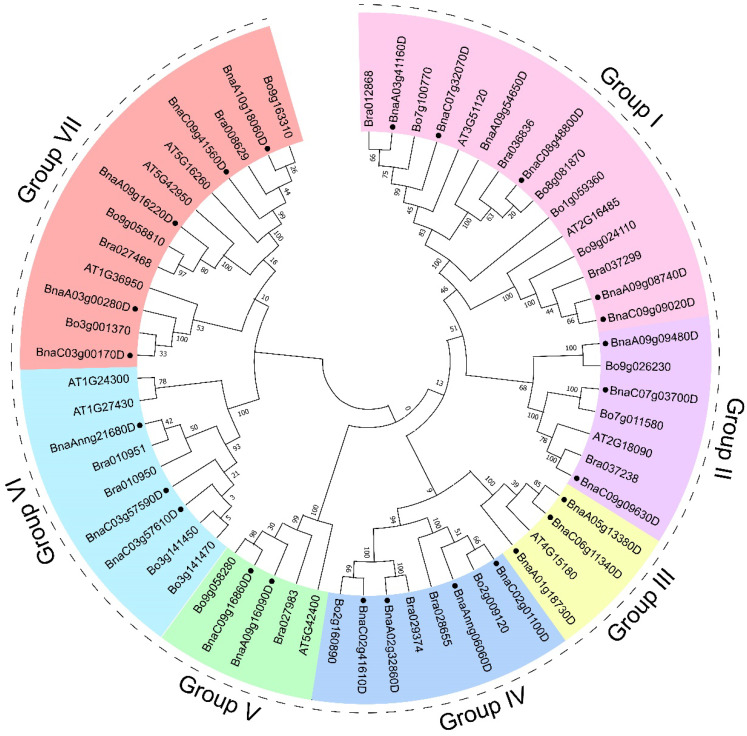
Phylogenetic tree of GYF proteins in *B. napus* (●), *B. rapa*, *B. oleracea*, and *A. thaliana*. The phylogenetic tree was generated by the NJ method with bootstrap analysis (5000 bootstrap replicates) using an amino acid sequence alignment of GYF proteins from *B. napus* (●), *B. rapa*, *B. oleracea*, and *A. thaliana* by the MEGA 11.0 program. GYF proteins were divided into 7 groups.

**Figure 5 genes-14-00224-f005:**
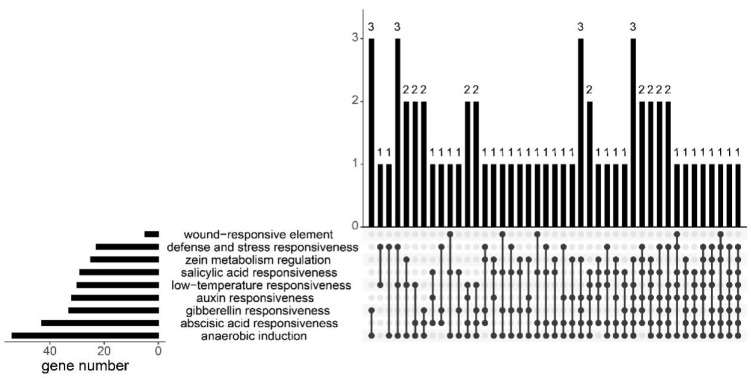
Distributions of cis-acting elements in promoters of *Brassica* GYFs and *AtGYFs*. The UpSet plot shows the distribution of cis-acting elements in *GYFs* promoters. The bar chart above represents the number of genes contained in each type of *GYFs*. The bar chart at the bottom left represents the number of genes included in each type of cis-acting element. The dotted line shows the types of cis-acting elements contained in the group.

**Figure 6 genes-14-00224-f006:**
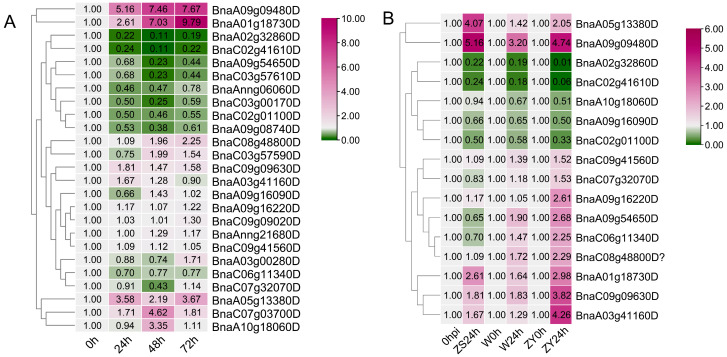
Gene expression heat map of *GYFs* in rapeseed cultivars induced by *Sclerotinia sclerotiorum*. (**A**) Gene expression heat map of ZHONGSHUANG11 (ZS11) infected by *S. sclerotiorum* at 24, 48, and 72 h post-inoculation. (**B**) Gene expression heat map of ZS11, Westar (W), and ZHONGYOU821 (ZY821) infected by *S. sclerotiorum* at 24 h post-inoculation.

## Data Availability

The datasets used and/or analyzed during the current study are available from the corresponding author on reasonable request.

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
