# Peer review of "Genome-Wide Identification of GYF-Domain Encoding Genes in Three Brassica Species and Their Expression Responding to Sclerotinia sclerotiorum in Brassica napus"

_genes, 2023, doi:10.3390/genes14010224_

Round 1

Reviewer 1 Report

1-Add the references for material and method parts.

2- As the author used the fold change method for gene expression analysis, add the reference and formula.

3- combine the results and discussion.

Author Response

Q1: Add the references for material and method parts.

Authors response: Thanks for your suggestions. We have been checked and added the references for material and method parts.

Q2: As the author used the fold change method for gene expression analysis, add the reference and formula.

Authors response: Thanks for your suggestions. Relative gene expression levels were analyzed using the 2-ΔΔCT method for this paper [ Livak and Schmittgen,2001].

Q3: combine the results and discussion.

Authors response: Thanks for your suggestions. We didn’t combine the results and discussion, for the following reasons:

The results section are simply and objectively reports what we found, without speculating on why we found these results; whereas the discussion section interprets the meaning of the results, puts them in context, and explains why they matter. In qualitative research, results and discussion are sometimes combined together. However, this study is a quantitative research, it’s of importance to separate the objective results from the interpretation of them. Thus, we separate the results from discussion according to the format of Genes.

Reviewer 2 Report

This article used genome wide identification of the GYF-domain encoding genes in three Brassica species and their expression responding to Sclerotinia sclerotiorum in Brassica napus. The study is well design and results are well presented. However, there are few questions which must be addressed.

The abstract is well presented but methods are not clearly mentioned, which methods are used and what specific results were achieved.

Line 53 to 62 the authors should provide brief details of economic and industrial importance of the rapeseed.

The losses and impacts of the S. sclerotiorum should be discussed in details. Also the mechanism and genetic pathway of the S. sclerotiorum on Brassica species must be discussed.

Also discuss diseases and host range of the S. sclerotiorum.

Section 2.2 and section 2.5 should be cited with relevant studies, respectively.

https://doi.org/10.1016/j.plaphy.2021.01.042, https://doi.org/10.1016/j.indcrop.2022.116090

details of section 2.6 must be provided in clear way.

Modify the conclusion by adding future recommendations and impacts of the current study.

Author Response

Q1: The abstract is well presented but methods are not clearly mentioned, which methods are used and what specific results were achieved.

Authors response: Thanks for your suggestions. We added methods and specific results at the proper site of the abstract.

Q2: Line 53 to 62 the authors should provide brief details of economic and industrial importance of the rapeseed.

Authors response: Thanks for your suggestions. We added detailed expatiation of the economic and industrial importance of the rapeseed at Line 54 to 59.

Q3: The losses and impacts of the S. sclerotiorum should be discussed in details. Also the mechanism and genetic pathway of the S. sclerotiorum on Brassica species must be discussed. Also discuss diseases and host range of the S. sclerotiorum

Authors response: Thanks for your suggestions. We discussed the losses and impacts of the S. sclerotiorum, also the host range of the S. sclerotiorum and the mechanism and genetic pathway of the S. sclerotiorum on Brassica species were discussed.

Q4: Section 2.2 and section 2.5 should be cited with relevant studies, respectively.

https://doi.org/10.1016/j.plaphy.2021.01.042, https://doi.org/10.1016/j.indcrop.2022.116090

details of section 2.6 must be provided in clear way.

Authors response: Thanks for your suggestions. We supplemented the details in the Materials and Methods section, and added several references. And the method for gene expression analysis was cited.

Q5: Modify the conclusion by adding future recommendations and impacts of the current study.

Authors response: Thanks for your suggestions. We modified the conclusion by adding a brief summary of the future recommendations and the impacts of the current study.